# Pan-Evo: The Evolution of Information and Biology’s Part in This

**DOI:** 10.3390/biology13070507

**Published:** 2024-07-08

**Authors:** William B. Sherwin

**Affiliations:** Evolution & Ecology Research Centre, School of Biological Earth and Environmental Science, UNSW-Sydney, Sydney, NSW 2052, Australia; w.sherwin@unsw.edu.au

**Keywords:** information, artificial intelligence, evolution, speciation, autocatalysis

## Abstract

**Simple Summary:**

I outline a unified framework to study the evolution of information across biological and non-biological realms. Will biology, including humans, benefit or suffer from new information developments, including artificial intelligence (AI)? First, are biology and information different? Life is an ordered form that reproduces, with variation, and any ordered arrangement, biological or not, potentially contains information. Ordered forms existed when Earth was just a ball of chemicals, and some chemicals reproduced. So, biological and non-biological information might be components of a unified process, ‘PanEvolution’ (‘Pan-Evo’), based on the same operations for living or non-living entities—innovation, transmission/replication, adaptation, and movement. This can produce separate groups: ‘species’. In current panspeciation, biological information—especially in our brains—is moving into its own environment. Harm to biology might be minimal if humans and AI both behave intelligently because vastly different environments are optimal for humans and AI-containing machines. This would be the first speciation-like event involving humans for millennia, but it will not be particularly hostile if humans learn how to evaluate information and cooperate to minimise the effect of both human stupidity and artificial simulated stupidity (ASS—failed AI). Integrated thinking about technological and biological evolution might lead to better developments.

**Abstract:**

Many people wonder whether biology, including humans, will benefit or experience harm from new developments in information such as artificial intelligence (AI). Here, it is proposed that biological and non-biological information might be components of a unified process, ‘Panevolution’ or ‘Pan-Evo’, based on four basic operations—innovation, transmission, adaptation, and movement. Pan-Evo contains many types of variable objects, from molecules to ecosystems. Biological innovation includes mutations and behavioural changes; non-biological innovation includes naturally occurring physical innovations and innovation in software. Replication is commonplace in and outside biology, including autocatalytic chemicals and autonomous software replication. Adaptation includes biological selection, autocatalytic chemicals, and ‘evolutionary programming’, which is used in AI. The extension of biological speciation to non-biological information creates a concept called ‘Panspeciation’. Panevolution might benefit or harm biology, but the harm might be minimal if AI and humans behave intelligently because humans and the machines in which an AI resides might split into vastly different environments that suit them. That is a possible example of Panspeciation and would be the first speciation event involving humans for thousands of years. This event will not be particularly hostile to humans if humans learn to evaluate information and cooperate better to minimise both human stupidity and artificial simulated stupidity (ASS—a failure of AI).

## 1. Introduction

Many people are wondering whether biology, including humans, will find benefit or harm in new developments in information, including artificial intelligence (AI) [1,2,3,4,5,6,7]. Such a question appears to identify biology and information as separate things. However, since the middle of the last century, it has been pointed out that biology frequently (some would say always) is based on various types of information [8,9,10,11,12,13,14,15,16,17,18,19]. Despite the pervasiveness of information in biology, biologists have been slow to incorporate approaches used for non-biological information or to recognise when information-related approaches are used frequently in biology, such as the most commonly used frequency-sensitive biodiversity index, Shannon [20]. This article discusses important parallels and connections, pointing out the possibility that biological and non-biological information might be components of a unified process of information evolution, which has been proceeding since before biology existed. This unified process could tentatively be named ‘Panevolution’ or ‘Pan-Evo’. Such naming would recognise that all information, inside or outside biology, is involved in the same four basic processes [19] (Figure 1):Innovation (Section 3.1 and Section 3.2).Transmission and replication, including the random processes therein (Section 4.1 and Section 4.2).Adaptation and processes such as selection that often produce adaptation (Section 5.1 and Section 5.2).Movement (Section 6.1 and Section 6.2).

A similar set of common processes has been proposed for ecology and evolution by Vellend [21] and for cultural evolution by Muthukrishna [22]. Often, several of these processes will be operating simultaneously at one or more levels, each of which may incorporate biological and/or non-biological aspects, such as molecules, individuals, populations, species, and ecosystems [21,23,24,25,26,27,28]. Of course, ecosystems include not only biology but all the physical aspects of the world. This article will first present ideas about information, life, and intelligence, then consider each of the four processes above, showing commonalities between biological and non-biological information. Next, there will be a discussion of how these four processes might result in speciation, both biological and non-biological. Finally, this article will examine how artificial intelligence could possibly benefit or harm biology, including humans, stressing that AI might not spell doom for humans if AI actually becomes intelligent, which is not certain, depending upon what we mean by ‘intelligent’ and how humans influence the development of AI, as discussed below. Some suggest that technological and biological evolution are already integrated and “selected by both intentional… and unintentional… factors” [29,30]. Thus, thinking about their coevolution in an integrated and intelligent fashion might lead to better developments. 

## 2. Information, Life, and Intelligence

Of course, one must set out what is meant by ‘information’ and ‘life’, and it turns out that both relate to order (versus disorder) and a related concept called ‘entropy’, and what is more, there is only a rather blurry distinction between ‘information’ and ‘life’. After discussing that, we will consider ‘intelligence’.

The important point of this article is that any ordered arrangement, whether biological or not, can be considered to potentially contain information. ‘Information’ is related to entropy, which can be thought of as the potential of any collection of objects or symbols to code a message. The title of Shannon’s paper [31], which many regard as the origin of information theory, makes it clear that a message must be communicated (“The mathematical theory of communication”). Thus, the measure he uses in that paper, derived from thermodynamics, is a measure of the potential for some assemblage to carry information, but its informativeness depends upon decoding by a recipient of some kind. In other words, informativeness depends not only upon what ordered arrangements can be made from the subunits but also on the availability of a person or system that can make use of the ordered arrangement—receive knowledge or change its state in some other way. Let us look at four groups of fourteen possibly informative letters and a measure of each group’s order or disorder, called Shannon entropy (in natural log scale) [31].

oooooooooooooo Entropy = zeroooooooohhhhhhh Entropy = 0.69oooyhaerlelhwu Entropy = 2.11hellohowareyou Entropy = 2.11

The first group has little potential to code information, and its entropy is low. The other groups have higher entropy and higher potential to code information. The group with the second lowest entropy could likely code only a small range of messages in English or any other system. The two highest entropy groups can potentially code many messages and contain the same letters in different orders. For a recipient who speaks English, the last group actually does encode a message, “hello how are you”, but that group has the same entropy as the other high-entropy group, which is just nonsense. Thus, it is not entirely satisfactory to take entropy as a measure of information because anything’s informativeness always depends upon that thing’s ability to be interpreted (and possibly used) by a recipient [8]. For the purposes of this article, ‘information’ will be taken to denote some kind of order that can potentially code for a meaning to an appropriate recipient, who has not necessarily been identified yet. 

Next, what is the definition of ‘life’? Often, this definition hinges on the maintenance and transmission of order—roughly the opposite of entropy. Again, there are many definitions, although many researchers consider that life is defined by the ability to use energy and materials to create and maintain some ordered form that can reproduce itself with variation [32]. However, it will be seen below that many things that are conventionally regarded as non-living have all these functions, often simultaneously. All previous definitions appear to be artificial and lack generality. So, is there a boundary between living and non-living, and if so, where is it? Viruses are often said to be on this boundary because although they can indeed use energy and materials to create and maintain some ordered form that can reproduce itself with variation, they can only do this within a host cell. That seems to be a very arbitrary decision because some other parasites can only do those things inside a host cell, including bacteria in the rickettsias or mycoplasmas. This article will use the only definition that comes close to representing all previous definitions, which is the one just offered: the “ability to use energy and materials to create and maintain some ordered form that can reproduce itself with variation”.

How do we apply these notions to Pan-Evo, which necessarily will contain a wide variety of types of objects—molecules, individuals, ecosystems, etc.—plus the variants within each category, such as isomers of molecules, different species in an ecosystem, or different variants within a species? Dealing with this array of possibilities requires definition and quantification of the units (e.g., individual) [28,33,34]. One approach to this definition comes from assembly theory; in this theory, each object can be characterised by the assembly index, which is the minimum number of steps needed to construct the object. In the past, it was proposed that assembly theory be used to distinguish biological and non-biological entities, but this is not widely accepted, and now the index and the number of copies for each type might be used to describe evolution in biology or outside it [35]. However, note that two very different types of object might have identical assembly indices, so it is necessary to add the relationships between the objects, as defined by whether there are common components in the assembly pathway and how many are shared [36]. Examples of pathways might be a molecule’s non-biological synthesis mechanism (natural or artificial) or part of a biological phylogenetic tree, as shown in Figure 2. In Figure 2, it can be seen that the number of steps is an inadequate summary of the differences between the entities produced. For example, in Figure 2a, starting from the same component, there are two steps in the assembly tree of A and also for C, but the intermediate component is not shared, so A and C would not be classified as being the same chemical simply because their assembly paths are the same length. Similarly, in Figure 2b, the number of steps might need to be adjusted by upweighting some steps that are chemically more challenging (transversions); also, the alternative pathways and back-mutation show that the number of steps is not a complete summary of the assembly tree.

Finally, what is ‘intelligence’, and could it occur inside or outside life? Obviously, we should not fall into the trap of assuming that human and machine intelligence will each remain the same in the future, so a generic definition is needed. There are currently qualitative and quantitative differences in the intelligence of machines and humans (and between different humans!). Throughout the last century, there have been many suggestions for how to formally quantify intelligence, first as ‘IQ’ [22,38], but more recently as ‘generalized intelligence, g’, which aims to quantify intelligence in a culture-free way; for example, twelve sub-tasks from an established general intelligence test map to the activity of similar parts of the brain [39]. Nevertheless, there is still an abundance of definitions of intelligence; this article will define intelligence as “the ability to acquire and apply knowledge and skills”, which comes close to representing all other definitions [40]. If we substitute ‘information’ for ‘knowledge’, we will see below that some living and non-living systems can do this. However, not all systems display intelligence, even in humans, as set out in a paper discussing ‘Natural Stupidity’ [4]. Some think it is likely that AI will never develop the same quality of intelligence that humans have, being already set on a course to become very ‘intelligent’ at some tasks but appalling at other tasks that we regard as simple [41]. This mimics the way that individual humans vary in their ability to display intelligence in different settings.

## 3. Innovation

### 3.1. Innovation in Biology

Biological variation derives from mutations in the DNA, with mechanisms ranging from single base (A,T,G,C) substitutions deletions or insertions to larger rearrangements, plus variation that does not involve base changes, called ‘epigenetic’, as well as changes that do not directly relate to DNA, such as behavioural variation [42,43]. Whether each of these is transmissible is discussed later. Secondarily, but usually much faster than mutation, innovation can occur through recombination: the exchange of information by physical breakage and reunion of the DNA string of information to unite variants that were previously on separate DNA molecules (or ‘haplotypes’). In relatively short stretches of DNA (e.g., a megabase), recombination rates are low, so haplotypes can persist and sometimes may have great adaptive significance [19].

Mutation models could also be employed as approximations for the production of novel variants at other biological levels. For example, at the ecological level, some models (e.g., single-nucleotide substitution or stepwise mutation) might be used as models of speciation that occurs by the alteration of a single character, such as the ‘magic traits’ discussed in the speciation literature [19,25,44]. On the other hand, models that assume that every new variant is unique (e.g., the infinite alleles model) might be more appropriate for species that occur via multiple changes that accumulate during a period when two parts of a single species’ range are separated by a barrier [19,25,44].

### 3.2. Innovation Outside of Biology

There are many examples of innovation in the physical world, such as the alteration of molecules by UV radiation [45]; assembly theory (above) enables us to categorise these. As well as naturally occurring physical innovations, human-guided innovation is obvious in software programming, etc. Moreover, many software programs are now being altered by allowing the system to make random alterations to its own information, deliberately mimicking biological mutation and recombination, as part of what is called ‘evolutionary programming’ and ‘neural networks’, discussed further below as part of non-biological adaptation [46]. There is plenty of evidence that AI is becoming more and more integral to innovation, though some suggest that AI may never be able to autonomously do what humans consider an invention (of course, AI may have its own standard of invention) [47]. Also, AI is free of many constraints of biological transmission and replication; for example, AI’s massive replication means that a bad change is not serious—such a change is easily weeded out by assessment of its (poor) performance—see Section 5.2.

## 4. Transmission and Replication

### 4.1. Transmission and Replication in Biology

In biology, many of the variants described above can be transmitted. Transmission might be to offspring through mechanisms such as inheritance of DNA or epigenetic modifications but also through mechanisms such as inheritance of a suitable nest site [43,48,49]. Moreover, the nervous system, although ultimately based on DNA code, can be modified by learning from non-relatives, which is the basis of much of cultural evolution [22]. There are three fundamental replication modes: the exponential type, seen with cells or individuals; the autocatalytic (hyperbolic) type, seen with some macromolecules; and the template-dependent (parabolic) type, as seen with nucleic acids [50]. Expression of these modes is often restricted, such as nucleic acids only replicating as a synchronous part of a cell replication cycle. Other molecules are partly independent of that cycle, including viruses, epigenetic modifications, and prions.

### 4.2. Transmission and Replication Outside Biology

The modes of transmission and replication just described also happen to ordered structures outside biology, especially by autocatalysis, which is when the product(s) of the reaction speed the reaction and thus catalyse their own synthesis (Figure 3 [45]). Autocatalysis can occur either by the molecule acting as a template or by other means, and of course, is dependent upon the availability of the raw materials. A huge range of molecular types and mixtures thereof can show autocatalysis [51]. Such non-biological replication can also involve more than one molecule. Intriguingly, such interactions may have been involved in the origin of the genetic code; aptamers are nucleic acids bound to specific target molecules, and one such aptamer includes the codon for arginine and preferentially targets binding to arginine [52].

Also, artificial intelligence is moving towards achieving fully autonomous replication, with no human involvement (except perhaps the provision of energy), plus the mutation and recombination discussed in the previous section. Electronic information can be programmed to use any of the replication modes above, and we see this replication everywhere, including in the computer malware that we all seek to avoid. For example, in generative adversarial networks, if one system has been devised or trained to make certain decisions (e.g., “this is/is not a picture of a bus”), another electronic system can, after slightly variable replication, be assessed by the second system on its ability to make the same decisions, and after many repetitions of this process, the second system can become equally good at making such decisions, though perhaps with some differences in the underlying code [53]. This rapid, massive process of trial–error–replication gives AI enormous power in many tasks.

## 5. Adaptation

### 5.1. Adaptation in Biology

Selection is the consequence of the fact that heritable variant entities that are better at converting resources into replicates inevitably become more numerous. In biology, there are copious examples of natural or artificial selection, such as adaptation to ambient temperatures. Often, selection results in organisms that are said to be optimally adapted to the environment within various constraints, such as the availability of appropriate variants for the selection to act upon, and tradeoffs between opposing selective pressures, such as mate choice versus the cost of the mating signal, the number of offspring versus the size of offspring, and the difficulty of adapting to seasonal or long-term fluctuations in the environment [54]. There is also non-heritable adaptation, such as behavioural avoidance of harm, including an excessively hot or cold temperature. In all adaptation, there are two types of information in play: the information available from the environment, such as the ambient temperature, plus the information in the organism, including non-heritable information, such as learned behaviour, as well as all heritable information, such as DNA sequence and some epigenetics. There is an interaction between the probability of various occurrences in the environment and the payoff expected if each occurs after an individual has taken one of several available courses of action, such as transmission of one of the two allelic versions of a gene that the individual possesses, or taking some evasive action. The result is quite complex, but in some cases, it is worth the individual’s effort to acquire more information about probability and occurrence because it can improve the outcome [55]. This can be addressed by a variety of entropic methods [19,56,57]. Such approaches are not confined to variation within species—a maximum entropy approach is also used for species assemblages [58].

### 5.2. Adaptation Outside Biology

Non-biological autocatalytic systems can show adaptation, in which there is competition between similar chemicals produced by factors such as radiation. Those that persist or reproduce better become more numerous. An example is the reaction shown in Figure 3, where an autocatalytic chemical can be altered by irradiation to make a new chemical whose autocatalysis outcompetes its ancestor chemical [45].

As seen above in the discussion of generative adversarial networks, some software development is now using a form of adaptation that accepts or rejects random alterations to the information in the code based upon autonomous training that, with the other three basic processes, leads to ‘evolutionary programming’ and ‘neural networks’ [46], including ‘Artificial Intelligence’ [59,60]. Hopefully, the information that is available for training will be as unbiased as possible, but bias could be provided deliberately or inadvertently by humans, who are not known for their impartiality, as set out in a paper discussing ‘Natural Stupidity’ [4]. For example, Google’s ‘AI Overviews’ tool has recently recommended adding glue to pizza and eating rocks [61]. One might hope that software engineers would be sufficiently logical to avoid such pitfalls, but it is worth noting that for one or two decades, a system of software engineers plus marketers gave us a major operating system in which, to stop it, you clicked on a button labelled ‘start’; there are many other such examples. Recently, there has been a plea for engineers designing electronic information systems to make greater use of long-standing methods by which biological systems handle information [62]. For example, it is expected that communications systems involving satellites and terrestrial transmitters would operate better if they used a type of selection amongst varying signals: transmission would be prioritised for signals with low signal-to-noise ratio [63].

Non-living systems that have some variants that are better at converting local resources into replicates might not only compete with other non-biological systems but also compete with living systems [6,7,64]. Whether this is likely will be discussed later.

## 6. Movement

### 6.1. Movement in Biology

Movement is commonplace in biology, including the heroic annual there-and-back migrations of some birds and butterflies [65], as well as the often much shorter lifetime dispersal from place of birth to place of breeding. Even apparently immobile species, such as plants, have extremely mobile life stages, such as seeds and pollen. 

In fact, movement directly affects biological information. Briefly, for any pair of locations, lower dispersal from a place of birth to a place of breeding, smaller population size, or greater time since separation will result in less sharing of molecular variants (‘alleles’ or ‘haplotypes’) between locations, so that knowing the allele type of an individual gives better information about that individual’s geographic origin, which is called greater ‘mutual information’ between variant identity and location of origin [16].

### 6.2. Movement Outside Biology

Outside biology, it is also common to see the movement of structures that show some order and thus have information-coding potential. This occurs at scales from molecular diffusion through winds and currents up to tectonic plate movements and astronomical processes. AI-related entities such as robotic soccer players, self-driving cars, and rovers can also move on the surface of the Moon or Mars. Of course, information can also move through means such as wires or radio transmissions without any physical movement of structures.

Reaction–diffusion models typically include the change in space and time of the concentration of one or more chemical substances due to the interplay of chemical transformations and diffusion; the usefulness of these models has been demonstrated in one-dimensional and two-dimensional experiments [66]. Reaction–diffusion models can also be useful descriptions of some processes involving biological information-carrying entities, such as the development of phenotypes [67], ecological invasions and epidemics [66], and transmission of information in nerve pulses [68].

## 7. Speciation

### 7.1. Speciation in Biology

Speciation is thought to often involve all of the four processes: innovation, transmission/replication, adaptation, and movement. Typically, there will be a generation of variant information via mutations or recombination, producing new combinations of existing variants. The frequency of these variants will be modified by stochastic processes and possibly by adaptive changes. At the same time, there may be movement to and from different environments, where different variants might be adaptive. The process is usually considered to take place over multiple generations [44]. But how we assess whether speciation has been achieved is subject to much debate; Mayr’s Biological Species Concept (also sometimes reproductive or isolation concept) is characterised as “groups of actually or potentially interbreeding natural populations, which are reproductively isolated from other such groups” [69]. Such species might also be called a group of individuals that preferentially affect each other’s heritable information by transmission between individuals, with less emphasis on affecting information by other methods such as competition, cooperation, or predation. There are many alternative definitions of a species, most of which attempt to focus on one or more of the logical consequences of the divisions envisaged by Mayr, such as discontinuity of the distributions of physical or genetic characteristics [44]. This avoidance of Mayr’s definition is because the diagnosis of species using that definition requires such an overwhelmingly large testing effort. Considering only eukaryotic species, there are potentially ~5 ± 3 million species [70]. If for each of those five million species we tested them against the nine most closely related potential species, then there would be tens or hundreds of millions of tests needed, with each test following two generations of hundreds of crosses between the potential species, plus a similar number of control crosses within potential species. This is clearly not achievable—a review found less than 1000 such investigations in eukaryotes [71]. Therefore, most species definitions attempt to avoid crosses and use some easier method to approximate Mayr’s idea of a species [44]. Also, some authors avoid the species concept altogether, preferring to discuss ‘taxa’, ‘evolutionary significant units’, or ‘management units’ [72,73].

It is worth pointing out that there is a similar difficulty in defining species outside the eukaryotes, where there are several other ways of facilitating or limiting the exchange of information between groups of microorganisms. Lan and Reeves [74] echo Mayr by saying that for microbial taxonomics, “a major factor in maintenance of species specificity… is the existence of a recombination barrier between species”, and then their article discusses many alternative methods of delineating microbial species, all of which are also frequently used in eukaryotes [74].

### 7.2. Speciation Outside Biology

It is always difficult to think about the future—it is said that in the 1940s, IBM’s president thought that in the long term, there would be a global market for a total of about five computers [75]. Moreover, at present, some of the most developed systems in information, large language models (LLMs, e.g., Chat-GPT), are still susceptible to hallucination, which, like human hallucination, is when the LLM produces text that is “nonsensical, or unfaithful to the provided source input” [76]. Nevertheless, we have consistently seen IT developments overcome barriers and outperform positive predictions. Thus, it is likely that non-biological information will evolve to become increasingly sophisticated at autonomous innovation, replication and transmission, adaptation, and movement. Given that these four functions underly all biological evolution, including biological speciation, it is worth asking how speciation-like processes might occur within the non-biological realm or between biology and non-biology.

How could we extend the concept of species beyond the biological realm? And what should we call such a process when defined both inside and outside biology? Note that the words ‘species’ and ‘speciation’ are not exclusive to biology, being used in chemistry, geology, etc., and there is a general need to identify the limits to the units that we are discussing [34]. However, perhaps a new name is needed to encompass the related processes that happen within and outside biology, and I suggest ‘Panspeciation’ and ‘Panspecies’. Rather than trying to extend each competing version of the species concept within biology, one might extend Mayr’s general species concept in the following way: that members of the same panspecies should be able to influence other members of the same panspecies more by exchange of information than by other processes such as competition, whereas members of different panspecies influence the information in members of other panspecies more by processes such as competition, cooperation, or predation rather than by the direct exchange of information. Of course, that does not completely preclude effects on information due to competition within species or transmission between species; it just evaluates their relative importance. As with the species definition within biology, the experimental work necessary to validate panspeciation might be inaccessible for some reason, in which case there are similar approaches to avoid that experimental work. For example, the idea of a panspecies might be related to the compatibility of particular groups of software and hardware, whether generated by humans or by AI [77]. Also, the panspecies definition might be based on the assembly theory mentioned previously, so a panspecies would be an ensemble of objects that have more common steps and components in their construction than they have with objects outside the panspecies (Figure 2).

## 8. Integration of Non-Biological and Biological

There have been some explicit attempts to integrate biological and non-biological information. Often, these rely upon the idea that certain systems of equations will give a common currency to be used across biological and non-biological evolution and ecology. Examples abound. One is the commonality of biological and automaton self-replication, including the importance of replication of the information [78,79]. Another example is the similarity of ‘drift’ in gas or electron mechanics to ‘selection’ in biology (despite the fact that the term ‘drift’ is used for a non-directional process in population genetics [80,81]). Another example is the similarity of Boltzmann’s thermodynamic equation and Shannon’s information equation [56], which is appealing because biological processes always have some underlying energetic basis [82]. However, simply showing that two equations have a similar form does not mean that they represent the same thing [83]. In a similar vein, there have also been attempts to unify the analysis of biological processes across scales from genes to ecosystems [21,24,25,26] and to integrate cultural and genetic inheritance using the Price equation [84]. Such integration might possibly be extended into non-biological systems with appropriate caution. 

In particular, O’Connor et al. [8] attempt to integrate the analysis of information, energy, and materials and suggest that biology is an emergent property of information processing systems at multiple scales. They stress that the behaviour of the system will depend upon the available information and that units of selection might extend beyond the biological individual.

An obvious point of commonality is the way that biological processes might have evolved out of non-biological processes. Autocatalysis appears from industrial application through to possible pre-biotic evolution [85]. Autocatalytic RNA molecules can undergo mutation, recombination, and selection [86]. Transfer-RNA activities are not just limited to protein translation but also have other functions, particularly in viruses where they can be involved in replication, reverse transcription, and telomeres (chromosome ends) [87]. Other non-RNA self-replicators are being used to build de novo life [88]; for example, catalytic DNA is more stable than RNA, so favoured industrially [89].

## 9. Potential Benefits and Harms

### 9.1. Possible Benefits to Biology, including Humans

There has been much recent discussion of artificial intelligence (AI) and whether it might cause a ‘singularity’, perhaps even human extinction [3]. Of course, if humans are actually intelligent (though see [4]), they will devise effective regulations to limit the societal harms of AI [1,2,90]. Irrespective of such regulation, AI might even help humans if AI actually becomes intelligent [29,30].

It has been pointed out that the interaction of heritable evolutionary changes and cultural changes might have given rise to humans’ extraordinary ability to cooperate with non-kin in huge groupings, leading to our present dominant status; in particular, the different transmission methods of genes and culture might have allowed such groups to maintain cultural cohesion without genetic relatedness [91]. Of course, everything in current information technology is part of human culture or a derivative thereof, so this symbiotic relationship might continue to prosper. If we accept that life is derived from information’s evolution but that we have so far mainly focused on understanding the evolution of living systems, we might see benefits in broadening our investigation to the evolution of all information, including the possible future.

Information changes in biology and the physical world might be managed for our benefit. An example of this would be to manage two evolutionary challenges: delaying adaptation of pathogens, pests, etc., and speeding adaptation of valued organisms [92]. However, note that this will require careful regulations because of the ever-present possibility of disagreement between individual and public good [92], as well as the possibility that AI might not help us but instead hinder us if AI is non-intelligent. Other possible benefits include the use of artificial replicators to build de novo life [85,88], the use of assembly theory for drug discovery [36], and information theory to predict the likely phenotypic effects of novel mutations before they are manifest in the next generation, either natural or in vitro [93]. Finally, information-based analyses can assist our analysis and forecasting of natural processes at many levels [19,46,56,94].

### 9.2. Possible Threats to Biology, including Humans

AI might pose several possible threats to biology, including humans and human–AI coevolution, as well as direct competition, which is discussed in the next section. An example of potential human–AI coevolution would be if AI takes over more tasks, such as your car identifying that its brakes need fixing and driving itself off to get repaired. (Hopefully, the AI will not decide that you want the cheapest brakes!) As a result, there might be selection against maintaining brain parts or functions that do things that AI can do for us. This selection against certain brain functions or regions might be driven by the energetic expense of running a large brain [95] and possible risks to mother and child during the birth of a baby with a large head [96,97]. Thus, what is initially a mutually beneficial arrangement might gradually result in increased dependency of humans upon AI, especially given the attention-getting strategies of AI [64]. However, note that currently undesirable allelic variants (e.g., for small brains) are usually recessive and rare, and therefore, it could take thousands of generations for such selection to change allele proportions very much. 

This article does not intend to extensively discuss what regulations are currently needed for AI, but there are some obvious ones, such as banning lethal AI, and a number of measures to increase trustworthiness in AI, including requiring each piece of information to have prominent identification of the humans or machines that generated it, which would limit malign influence in politics or elsewhere [41]. This regulation will, of course, be anthropocentric and will only be achievable if humans themselves behave intelligently [4]. Thus, there is a need to educate humans to better understand how we should be evaluating information and its sources, as well as improving cooperation. This will minimise the effect of human stupidity [4] and avoid producing artificial simulated stupidity (ASS). We are, or can be, in charge of this process and need to act [22].

## 10. AI, Competition, and Panspeciation

In biological evolution, distinct lineages often evolve to occupy different environments [44,98]. Note that humans and machines are best suited to vastly different environments (Figure 4). Humans require things that damage computers and other machines: copious water, oxygen, and carbon dioxide for their food plants. Also, machines can be manufactured to tolerate a much wider range of temperatures and radiation than humans. Thus, environments most favourable to humans are present on much of Earth, whereas environments more favourable to machines are scarce on Earth but abundant elsewhere, such as the Moon and Mars. So, if AI is actually intelligent, competition might be minimised because machines might principally seek different environments than humans [5]. Of course, we might continue other human/machine interactions that do not need physical proximity [29,30].

As machines improve at operating their own design and production, the information that previously resided in humans’ minds might diverge into two groups with the independent transmission of information: some still transmitted by humans and some entirely independently transmitted—would we call that ‘incipient panspeciation’? As stated above, we often say that two groups belong to different species if their internal interactions have a greater emphasis on the transmission of information, whereas between-group interactions have a greater emphasis on competition, cooperation, or predation. Diagnosis of separate species status is especially likely if the groups also display environmental segregation [44,98], as just postulated. If this comes to pass, then humans and machines might show relatively little competition because of their different favoured environments (Figure 4). Thus, spatially separated coexistence might be possible, depending on the variation in humans and AI—both the differentiation between the average members of each group and the variation within each group, which also affects competition and coexistence [99,100].

## 11. Conclusions

Obviously, one could completely ignore the similarities between biological and non-biological evolution, but the processes outlined in these two arenas have been proceeding for millennia and will likely continue to do so. Considering their combined evolution as Panevolution, including panspeciation, may lead us to better understand and manage the current processes. If someone might like to use the word ‘panspeciation’ to describe the splitting of information into that carried by humans and that carried by AI and its successors, then we are currently in the early stages of the first speciation-like event involving *Homo sapiens* for many thousands of years. But it would not be particularly hostile to humans if AI is actually intelligent and pursues its own best interests, which probably do not involve severe competition with humans.

At present, the only likely source of artificial intelligence is from human-run brains and organisations. For various reasons, this situation may not last, but it means that we currently have the capacity to limit any harm if we wish to do so. In particular, it will be important to be quite sure of what we mean by ‘intelligence’ and make sure that it is possessed not only by AI but also by those humans who manage AI so that both can see the benefits of partitioning the environment rather than competing. To minimise human stupidity and artificial simulated stupidity (ASS, a failure of AI), humans need to be educated to evaluate information and insist on the identification of sources, to limit malign influences. Integrated thinking about technological and biological evolution might lead to better developments.

## Figures and Tables

**Figure 1 biology-13-00507-f001:**
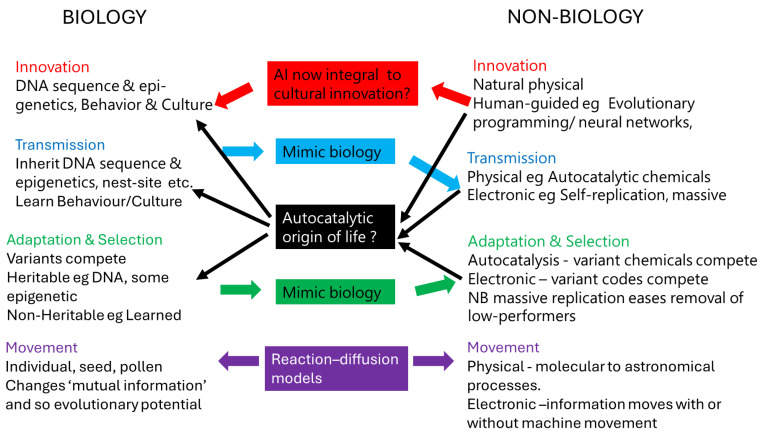
The four basic processes in Panevolution. Interactions between biological information and non-biological information are shown in the centre column. Coloured arrows indicate some ways in which methods for handling information have transferred from one field to another in the last century or so, while the black arrows show some of the ways in which natural information processes might have moved from non-biological processes to biological processes during the origin of life. Further details are in Section 3.1, Section 3.2, Section 4.1, Section 4.2, Section 5.1, Section 5.2, Section 6.1 and Section 6.2.

**Figure 2 biology-13-00507-f002:**
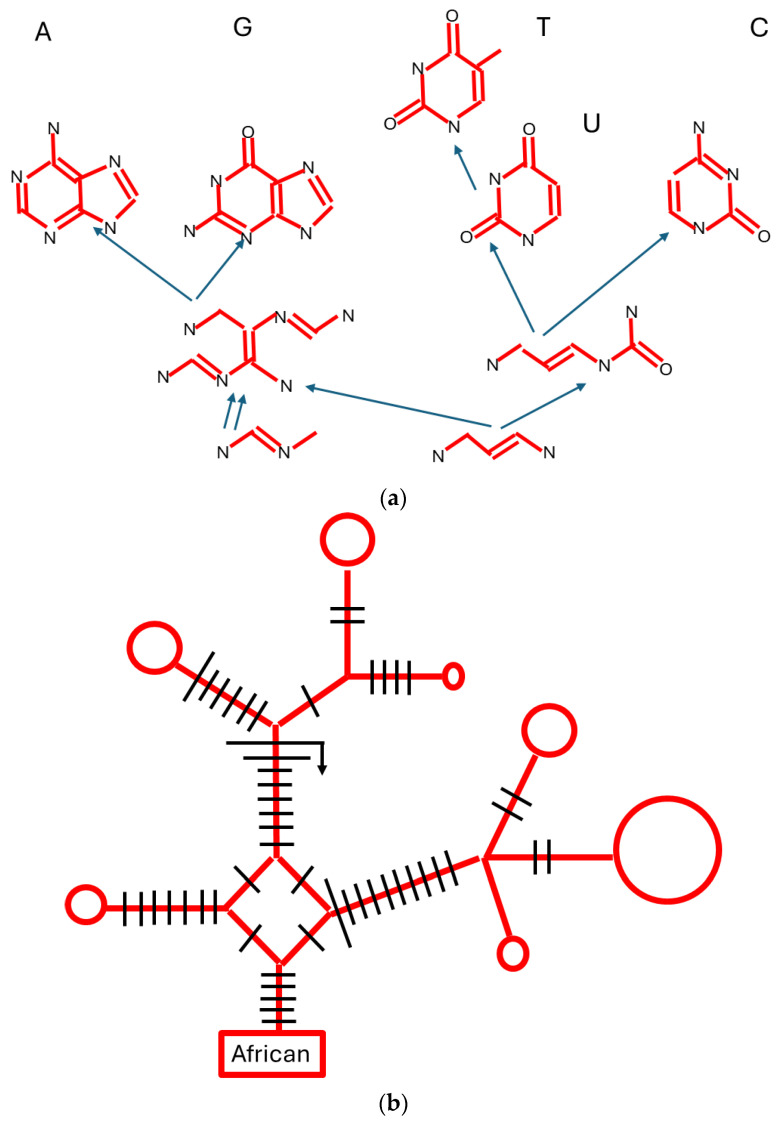
Assembly trees, non-biological and biological. (**a**) A non-biological partial molecular assembly tree for synthesis of the nucleic acid bases; purines A and G; and pyrimidines T, U, and C. The letters N and O denote oxygen and nitrogen, and vertices without a letter are carbons (redrafted from [36]). (**b**) A biological phylogenetic network of Finnish human mitochondrial DNA sequences, originating from African ‘root’. Size of circles is proportional to the number of identical sequences at that node. Bars indicate base alterations that are all transition substitutions (pyrimidine for pyrimidine, T↔C or purine for purine A↔G), except for the intermediate-length bars, which are the more chemically difficult transversions (purine to pyrimidine, or vice versa). The very long bar with an arrow indicates a back mutation to a state present earlier in the phylogeny. Notice the diamond shape in the centre of the network that shows alternative substitution pathways in the origin of the sequences above it. This is a partial redraft of the network in [37].

**Figure 3 biology-13-00507-f003:**
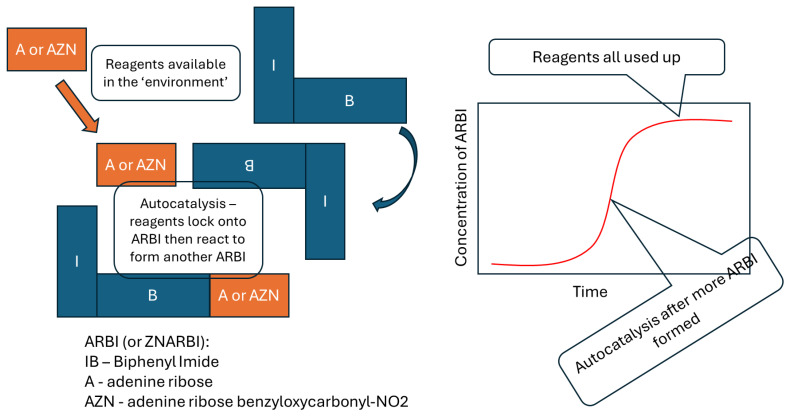
An autocatalytic reaction. Similar reactions happen for adenine ribose biphenyl imide (ARBI) or benzyloxycarbonyl-NO2 adenine ribose biphenyl imide (ZNARBI). If the environment includes radiation, then ZNARBI is mutated to ARBI, and ARBI replicates faster than ZNARBI, so ARBI outcompetes or is more ‘adapted’ to that environment. Simplified from [45].

**Figure 4 biology-13-00507-f004:**
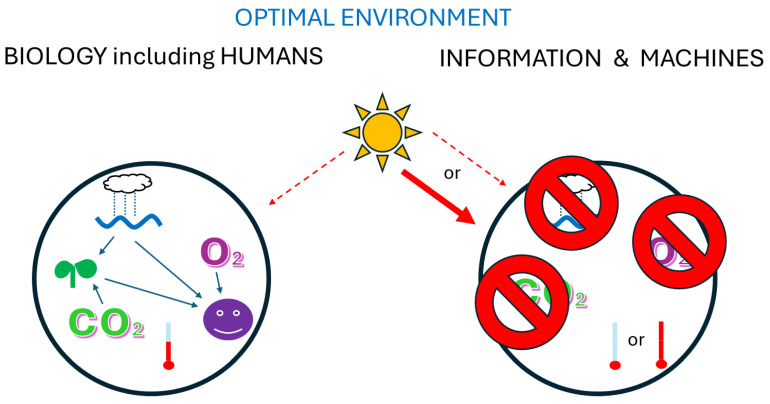
Suitable environments for information held in biology or outside of biology. Biology requires water, oxygen, carbon dioxide, moderate temperatures, and relatively low cosmic radiation. The machines that carry information outside of biology are damaged by water, oxygen, and carbon dioxide and can be manufactured to tolerate a much wider range of temperatures and radiation.

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
