# Peer review of "Pan-Evo: The Evolution of Information and Biology’s Part in This"

_biology, 2024, doi:10.3390/biology13070507_

Round 1

Reviewer 1 Report

Comments and Suggestions for Authors

I think that the abstract is missing in it’s flow. There are some definitions mentioned directly without a link between sentences. Also, a brief sentence on what this review covers or what it highlights could have been mentioned in the abstract. Maybe the research gap could be more descriptive in the introduction. The conclusion can be rewritten and can be described more concluding the highlights of this manuscript review. May be what are the ways to minimize artificial simulated stupidity or human stupidity can be included like awareness and open communication, etc. What are alternative theories to pan speciation could be included. Overall, very interesting manuscript review. 

Reviewer 2 Report

Comments and Suggestions for Authors

Summary

The author in this manuscript interconnect information, life, and intelligence as concepts centered around order and entropy. Information involves ordered arrangements that can encode messages, with entropy measuring the potential to convey information. Life is defined by the ability to maintain and transmit order, utilizing energy to reproduce with variation, often blurring the line between living and non-living systems, such as viruses. Intelligence, the ability to acquire and apply knowledge, extends to both biological and non-biological systems, including AI. Innovations occur through genetic mutations and recombination in biology and evolutionary programming in AI, with replication mechanisms seen in both biological inheritance and non-biological processes like autocatalysis and AI replication, demonstrating a continuum between living and non-living entities.

Strengths

The idea of integrating biological and non-biological information systems is innovative. The concept of panspeciation, where species-like groups are defined based on shared information processes, bridges biological evolution with artificial intelligence and other information systems. This offers a novel perspective on how we might understand and categorize entities beyond traditional biological definitions. The emphasis on how movement affects biological information and genetic variation is also significant. By highlighting how dispersal impacts the sharing of molecular variants, the paper underscores the importance of spatial factors in genetic studies and evolutionary biology. Applying reaction-diffusion models, typically used in chemistry, to biological processes like phenotype development and nerve impulse transmission, showcases a cross-disciplinary approach that enriches both fields. This methodological crossover demonstrates the potential for chemical models to provide insights into biological systems. The discussion on speciation focusing on information transmission rather than traditional genetic or reproductive isolation is a fresh perspective. This approach could lead to new ways of studying and understanding species formation and evolutionary processes. The recognition of the difficulties in defining species both in microorganisms and eukaryotes, and the comparison of various methods to address these challenges, provides a comprehensive overview that is crucial for taxonomic and evolutionary studies.

Specific Gap Addressed

The primary gap this paper addresses is the lack of a unified framework to study the evolution of information across biological and non-biological realms. By proposing the concept of panspeciation, the author suggests a novel way to understand and categorize information-processing entities, bridging a significant gap between biological evolution and artificial intelligence. Another gap addressed is the understanding of how movement influences genetic variation and information sharing within populations. The manuscript emphasizes the importance of spatial dynamics in genetic studies, which is often underappreciated in traditional evolutionary biology. The rethinking of speciation in terms of information transmission rather than just genetic or reproductive isolation offers a new dimension to evolutionary biology. This perspective addresses the gap in understanding species formation in a way that could integrate technological and biological evolution.

Weakness

While the interconnectedness of information, life, and intelligence provides a broad conceptual framework, the explanation presented has several limitations and ambiguities. The definition of information hinges too heavily on entropy, which doesn't always capture the practical utility or meaningfulness of data. Life's definition, tied to order and reproduction, fails to address the complexity and nuance of living systems, especially with entities like viruses that straddle the boundary between living and non-living. The discussion of intelligence, particularly in AI, oversimplifies the depth and scope of human cognitive abilities versus machine learning capabilities, neglecting the qualitative differences in understanding and context. Overall, the manuscript pushes the boundaries of traditional biological studies by proposing interdisciplinary methodologies and new conceptual frameworks that can be applied to both biological and artificial systems.

Reviewer 3 Report

Comments and Suggestions for Authors

Review of: Pan-Evo: The evolution of information, and biology’s part in this.

The paper offers an insightful interpretation about the possible outcome of the growing interaction between biology and non-biological information. After offering an interpretation of the concepts of information, life and intelligence, it shows how these might interact in four different ways in the biological and non-biological world, especially AI. It then suggests that the interaction might result in a pan-speciation event that would lead humans and AI machines to occupy different environment and that would not necessarily be negative for humans.

The paper surely sheds an original light on the much debated subject of opportunities and dangers of AI systems, and the comparison between the working and evolution of biological and non-biological systems is very interesting. Its structure is clear and the paragraphs are short and to the point, supported by relevant examples and sound bibliography. So, it is surely worth publishing.

This being said, I think it would gain in clarity with some minor modifications and improvements:

·      Line 53-58 and figure 1: The meaning of each of these processes (innovation, transmission…) should be anticipated here and explained a bit more in detail, also with some example. Also, figure 1 should be better explained (including the meaning of the arrows and of their directions), either in the text or in the figure’s caption.

·      Line 64-70: I would develop a bit more this summary of the content and also add the number of the sections where each idea is developed. Also, the thesis of the article should be clearly explained at this point.

·      Section 2: this section talks about information, life and intelligence. While the discussion about information is well structured and end with a proposal of a definition to be used in the paper (line 107-109: “For the purpose of this article…”), discussions about life and intelligence are a bit confused and superficial. Regarding ‘life’, it is not clear to me why the author moves from discussing life to discussing units; and the question of defining life is left unanswered. Regarding ‘intelligence’, a too short and superficial discussion is offered, and we do not get any definition of intelligence to be applied in the article. I suggest to review the structure of this section and to provide some definition of life and intelligence that will be used in the paper.

The rest of the article is clear and to the point, although I would develop a bit more the conclusions (including restating the initial thesis and showing how it has been demonstrated thorough the article). I have the general feeling that too little is concluded after so much is said and so much material is offered. For instance, one might conclude that the real subject of evolution is not life, but intelligence/information, and that living entities are just the first step of this process, now integrated by non-biological AI entities, and who knows what other kinds of entities in the future: they are and will be just the vehicles for this evolution. 

Some general comments:

·      The title is unfortunate, as the paper covers much more than just the evolution of information. I suggest choosing a more appealing one.

·      Lines 84-86: the structure of the sentence sounds a bit weird. Consider rephrasing: “The title of Shannon’s paper, that many…, makes it clear…”

·      Section 3.b – line 193-195: “There are many examples of innovation outside of the physical world…”: provide some or at least provide some reference.

·      An interesting proposal to unify the mechanisms of evolution in biological and cultural systems can be found in : Baravalle, Lorenzo & Luque, Victor J. (2022). Towards a Pricean foundation for cultural evolutionary theory. Theoria 37 (2):209-231.
